# Towards Generalizable 3D Human Pose Estimation via Ensembles on Flat Loss Landscapes

**Jumin Han**
Department of Artificial Intelligence
Korea University, Seoul, South Korea
juminhan@korea.ac.kr

**Jun-Hee Kim**
Department of Artificial Intelligence
Korea University, Seoul, South Korea
jh__kim@korea.ac.kr

**Seong-Whan Lee**
Department of Artificial Intelligence
Korea University, Seoul, South Korea
sw.lee@korea.ac.kr

## Abstract

3D Human Pose Estimation (HPE) is a fundamental task in the computer vision. Generalization in 3D HPE task is crucial due to the need for robustness across diverse environments and datasets. Existing methods often focus on learning relationships between joints to enhance the generalization capability, but the role of the loss landscape, which is closely tied to generalization, remains underexplored. In this paper, we empirically visualize the loss landscape of the 3D HPE task, revealing its complexity and the challenges it poses for optimization. To address this, we first introduce a simple adaptive scaling mechanism that smooths the loss landscape. We further observe that different solutions on this smoothed loss landscape exhibit varying generalization behaviors. Based on this insight, we propose an efficient ensemble approach that combines diverse solutions on the smooth loss landscape induced by our adaptive scaling mechanism. Extensive experimental results demonstrate that our approach improves the generalization capability of 3D HPE models, and can be easily applied, regardless of model architecture, with consistent performance gains.

## 1 Introduction

3D Human Pose Estimation (HPE) is one of the essential computer vision tasks. In recent years, it has received increased attention due to its successful application in the areas of autonomous driving [33, 39], robotics [40, 12, 26], and industrial safety [31] along with the advancement of Deep Neural Networks (DNNs). Especially, in the domain of industrial safety, it is crucial to ensure the generalization capability of the model as its accuracy can be directly tied to safety concerns.

Unfortunately, it is challenge to get a robust 3D HPE model due to the inherent complexity and variability of human poses across different datasets and domains. Previous researches have addressed these challenges by focusing on multi-hypothesis concept [8, 4, 22, 19], learning better posture information [16, 10, 29], designing elastic model architectures [36, 27], and data augmentation [7, 28]. Although previous methods have achieved notable performance improvements, they often overlook the structure of the loss landscape, which plays a critical role in understanding model generalization and stability [5, 1].

In our analysis, we found that the loss landscape of 3D HPE task can be highly complicated and has multiple disconnected local minima. Such disconnected modes can pose significant challenges for

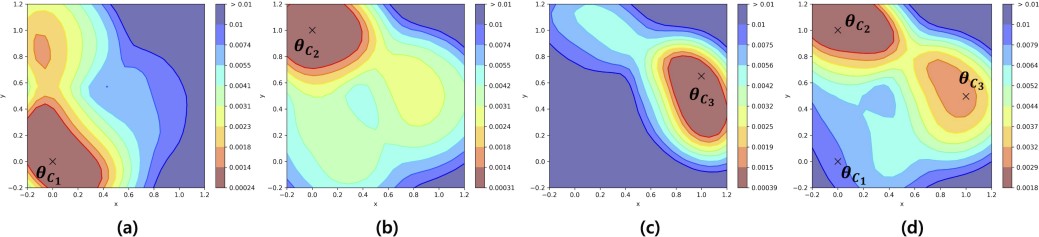

Figure 1: The local and global loss landscape when $K$ is 3. (a) Local loss landscape of $C_1$. (b) Local loss landscape of $C_2$. (c) Local loss landscape of $C_3$. (d) Global loss landscape of $\mathcal{D}$. The local loss landscapes have various shape depending on the degree of depth ambiguity. Especially, the global loss landscape and some local loss landscapes have multiple local minima. $\theta_{C_1}$, $\theta_{C_2}$, and $\theta_{C_3}$ are model parameter around local minimum of each local loss landscape. Note that the local loss landscape of $C_k$ is a result when the model trained with only $C_k$.

optimization and may adversely affect the generalization performance of the model (*i.e.* converging to the sharp minimum). Despite its critical impact on generalization, there has been lack of attempt to design training or inference strategies that explicitly leverage this insight in the 3D HPE field. We argue that this structural insight is critical to enhance the generalization ability of the model because simply optimizing a single model over such loss landscape may result in a biased convergence toward sharp minimum, which is not desirable.

To mitigate this, we propose a simple adaptive scaling mechanism to smooth the loss landscape, effectively reducing the likelihood of existence of high-loss barriers between local minima and forming a loss landscape geometry where diverse solutions reside on a low-loss region. Furthermore, we found that the diverse solutions on the smoothed loss landscape by our adaptive scaling mechanism exhibits various generalization behavior. Based on this observation, we also introduce an efficient ensemble strategies with the adaptive scaling mechanism that combine multiple solutions on the smooth loss landscape without destructive interference to improve robustness and generalization ability of the model. It is worth noting that our method requires a similar amount of time as training a single model and can be applied regardless of the model architecture.

Our contributions are as follows:

- We propose a novel and simple ensemble method that can be integrated with simple modifications, regardless of the model structure. The proposed method is designed to induce flatness of loss landscape while concurrently enable parameter-efficient ensemble method, and can be implemented easily.

- We shed light on the difficult learning process of 3D HPE task through a analysis from a loss landscape perspective. To the best of our knowledge, we are the first to take this direction of analysis in a 3D HPE task.

- Our method enhances performances of the model for the representative model architectures (MLP, CNN, GCN, and Transformer) of 3D HPE in benchmark datasets such as Human3.6M [14], MPI-INF-3DHP [24], 3DPW [30], and BEDLAM [2].

## 2 The Loss Landscape of 3D Human Pose Estimation

In this section, we explore the loss landscape of 3D HPE task. During our experiment, we leverage the dataset, Human3.6M [14], which is popularly utilized as benchmark dataset in the 3D HPE task. For the network, we utilized the BaselineNet from a foundation work [23]. For the results of other network architectures, see the Appendix. We start our analysis from elaborating the experimental setting and ending with summary of our analysis and key insight.

### 2.1 Experimental Setting

Considering the loss value for a whole dataset $\mathcal{D}$ at parameter $\theta$ is a linear combination of loss values for individual examples in $\mathcal{D}$ at $\theta$, analyzing loss landscapes for subsets of $\mathcal{D}$ is an effective approach to understand the loss landscape. To efficiently explore the high-dimensional loss landscape of 3D

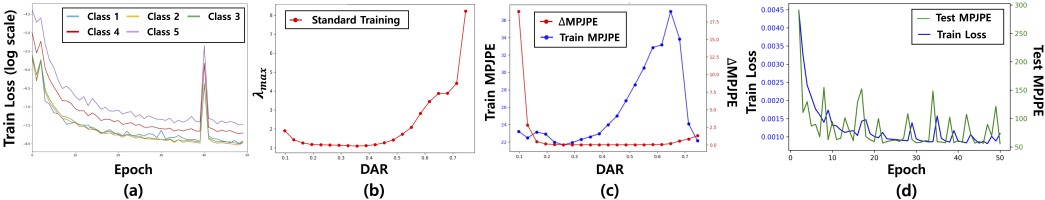

Figure 2: The different loss landscape characteristics depending on each class. (a) Class-wise training loss per epoch when $K$ is 5. (b) Top-1 eigenvalue of each local loss landscape when $K$ is 21. (c) Class-wise original train MPJPE (blue line) and the gap between MPJPE after small input perturbation and original MPJPE (red line). Note that the (b) and (c) are results when the model is trained with the whole dataset $\mathcal{D}$ when $K$ is 21. (d) The training trajectory of the model when the model is trained with $D \backslash C_{20}$ when $K$ is 20 (blue line is train loss and green line is test accuracy).

HPE, we first divided the dataset according to the degree of ambiguity of each example, which can be different depending on inherent data patterns, rather than random partitioning. The inherent depth ambiguity of 3D HPE can be understood as a one-to-many mapping between the input and output space, which means multiple valid loss values can exist for a single input. Especially, a high degree of depth ambiguity implies a high variance of the valid loss values for a single input. Since this property directly affects the geometry of the loss landscape, partitioning based on depth ambiguity is an appropriate choice. To quantify of the depth ambiguity per example, we borrow a definition, Depth Ambiguity Ratio (DAR), from a recent work [16]. With this DAR, we can add the quantified depth ambiguity to each data pair. The train dataset can be re-expressed as $\mathcal{D} = \{(x_i, y_i, u_i)\}_{i=1}^{N}$. The $x, y$ are the coordinates of 2D pose and 3D pose, respectively. The $u$ is the DAR for a data pair $(x, y)$ and $N$ is the size of dataset. To partition the dataset based on the values of $u$, we sorted $\{u_i\}_{i=1}^{N}$ in ascending order and defined the bin edges $\{b_k\}_{k=0}^{K}$ by selecting values from the sorted $u$ values such that $b_0 < b_1 < \cdots < b_K$. Each example $(x_i, y_i, u_i)$ is assigned to the $k$-th bin if it satisfies $b_{k-1} \leq u_i < b_k$ for $k = 1, 2, \ldots, K-1$, and $b_{K-1} \leq u_i \leq b_K$ for $k = K$. See Appendix for the detailed binning procedure. From now on, we call the subset assigned to the $k$-th bin as a class $C_k$. So, the degree of the depth ambiguity for each class can be ordered as $C_1 < C_2 < \cdots < C_K$. For convenience, we call the loss (landscape) of each class as local loss (landscape) and the loss (landscape) of whole dataset as global loss (landscape) from now on.

## 2.2 Local Loss Landscape of Each Class

Considering that the global loss is a linear combination of the local losses, analyzing each local loss landscape significantly aids in understanding the global loss landscape. To this end, we begin by analyzing the individual local loss landscapes. At first, we present a simple results that the training loss roughly varies depending on each class in Figure 2(a). There can be various reasons for this, the main factor of this would be the different local loss landscape for each class. In this reason, we explore the top-1 eigenvalue $\lambda_{max}$ of the Hessian matrix for each class to see the local loss landscape curvature around the trained model parameter with $\mathcal{D}$ in Figure 2(b)[1]. As expected, the curvature of local loss landscapes are different depending on each class. While the $\lambda_{max}$ of local loss landscapes for the both extremely low and middle degrees of depth ambiguities are similar, their geometries are different to each other. To show this, we inspect the loss value after input perturbation for each class in Figure 2(c)[2]. The data for extremely low degree of depth ambiguity exhibited a tendency to be memorized by the model because the model cannot appropriately address small input perturbation. Considering the extremely low degree of depth ambiguity as one-to-one mapping relationship between input space and output space, this is natural. On the other hand, the data for high degree of depth ambiguity exhibited steep loss landscape, which implies that they are not being learned enough and the model is located on the unstable loss landscape. These all results indicate that the local loss landscape for each class is shaped differently, suggesting that the global loss landscape can be highly complicated depending on the individual local loss landscape. To show this empirically, we train the network with $D \backslash C_{20}$ when $K$ is 20. We can observe that the training trajectory is highly complicated when we see the training loss and test accuracy through epochs in Figure 2(d). This

---

[1] We estimate the top-1 eigenvalue $\lambda_{max}$ of each local loss landscape following [34].

[2] Similar to [20], we perturb the input 2D pose in a scale-invariant manner with Gaussian noise. The perturbed input with a perturbation scale $s$ is expressed as $x_{pert} = x + \frac{d}{\|d\|} \cdot \|x\| \cdot s$ where the $d \sim N(0, I)$.

indicates that a small change of arbitrary local loss landscape may lead to highly complicated global loss landscape.

## 2.3 Visualizing Loss Landscapes

In this section, we visualized the local and global loss landscape of 3D HPE task to inspect the local loss landscape in detail[3]. As expected, the local minimum of each local loss landscape are located in the different parameter space as shown in Figure 1. As we mentioned above, the global loss is a linear combination of local losses, thus leading to the highly complicated global loss landscape that potentially has disconnected multiple local minima, which cannot be distinguished from the perspective of model. This is because the model cannot distinguish which one is better because the gradient at those local minima are all same as zero. In fact, even when the $K$ is 3, the global loss landscape has two local minima as shown in Figure 1(d), which implies that the model can converge to different local minimum depending on their initial parameter or some arbitrary dominant local loss landscape, or even cannot be converged without delicate training strategy. For the results of various $K$ values, see Appendix. These disconnected local minima can exhibit different generalization behavior depending on their own sharpness and the loss value at the minimum[5, 1, 32].

## 2.4 Summary and Key Insight

As shown in Section 2.3, the local minimum of each local loss landscape can be differently located in the parameter space and they are all potential local minima in the global loss landscape. If the local minima of global loss landscape are disconnected to each other and their loss level are different at their local minimum, the model tends to be overfitted to an arbitrary dominant class $C_k$ so that generalization capacity can be deteriorated or exhibits highly unstable training process. Hence, it is important to reduce the likelihood that disconnected local minima may exist in the global loss landscape. This consideration leads us to our key insight to mitigate this as follows:

- Flatness of local minimum of each *local* loss landscape.

If the local minimum of each local loss landscape is sufficiently flat, there will be no significant high loss barriers between them, which reduces the likelihood of multiple disconnected local minima in the global loss landscape. We empirically validated this intuition in Section 4. This property facilitates stable training of 3D HPE model and may improve the generalization capability of the model. Based on this, we introduce a simple method motivated by this insight in the following section.

## 3 Proposed Method

Motivated by our key insight in Section 2, we aim to mitigate the problem of highly complicated loss landscape of 3D HPE task. To this end, we first propose a simple yet effective adaptive scaling mechanism that smooths the loss landscape of 3D HPE model. We elaborate our detailed method in following sections.

## 3.1 Adaptive Scale Adjustment of Prediction

In general, the behavior and output of 3D HPE network can be expressed simply by the following equation, where the network predicts the 3D pose $\hat{y}$.

$$\hat{y} = f_\theta(g_\phi(x)), \tag{1}$$

where $x$ is input 2D pose, $g_\phi$ is the network before last layer, and $f_\theta$ is last layer. Our core idea is to encourage flatness in the local minimum of the local loss landscape. Since the meaningful maximum value of $K$ is the dataset size, in the case where the number of class $K = |D|$, this is equivalent to promoting flatness around the local minimum of local loss landscape for every individual training example. One simple method for this is to apply adaptive scaling mechanism to Eq. 1 as follows.

$$\tilde{y} = \frac{f_\theta(g_\phi(x))}{\sigma(h_\psi(g_\phi(x))) + 1}, \tag{2}$$

---

[3]We visualized the loss landscape following [6]. For batch normalization [13], we run one additional forward pass for one epoch to calculate the running mean and standard deviation of activations of each layer with interpolated network weights.

where $h_\psi : \mathbb{R}^d \to \mathbb{R}$ that predicts a scalar value and $\sigma$ is ReLU function. The $d$ is output dimension of $g_\phi$. This simple mechanism induces the flatness of entire loss landscape.

For example, the $\hat{y}$ in Eq. 1 can be expressed as $a\hat{y} \times \frac{1}{a}$ where $a$ is a scalar value. This is similar form of Eq. 2 where $a\hat{y} = f_\theta(g_\phi(x))$ and $a = \sigma(h_\psi(g_\phi(x))) + 1$. Then for a overall network parameter set of Eq. 2 $S = \{\theta, \phi, \psi\}$, this means that there exists multiple solutions such that $\{S \mid \frac{f_\theta(g_\phi(x))}{\sigma(h_\psi(g_\phi(x)))+1} = \tilde{y}\}$ that predict the same $\tilde{y}$ in Eq. 2 depending on the value of $\sigma(h_\psi(g_\phi(x))) + 1$. This redundancy creates a smooth region in the loss landscape, which is known to correlate with better generalization. The empirical validation of this is provided in Section 4. Also, this simple mechanism enhances expressive power of the model because the scale parameter $\sigma(h_\psi(g_\phi(x)))+1$ is input-dependent, thus the model is capable of representing more diverse functions that could not be expressed before.

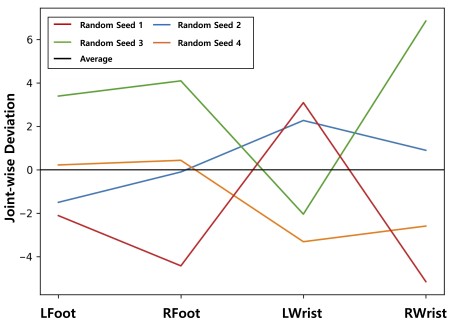

The adaptive scaling mechanism allows for various parameter solutions by intentionally introducing redundancy into the loss landscape. This naturally raises an important question: how do these distinct solutions differ in their generalization behavior? To explore this, we conducted a simple yet illustrative experiment to examine the generalization behaviors of models trained under the adaptive scaling scheme with different random seeds. Interestingly, we observed that the resulting models from different random seeds exhibited diverse generalization behaviors, as shown in Figure 3. We calculate the deviation of model predictions for joints that are difficult to estimate. For example, the deviation of $j$-th joint for $i$-th model can be calculated as $p_j^i - \mu_j$ where $\mu_j$ is $\frac{1}{n}\sum_{i=1}^n p_j^i$ and $p_j^i$ is a $j$-th joint of prediction from model trained with random seed $i$. This observation

Figure 3: Deviation of predictions from each solution on the smooth loss landscape for difficult joints of 3DHP [24]. Note that the models are trained with H36M [14].

led to our second key insight: ensembling these solutions can effectively leverage this diversity to further improve generalization capability.

## 3.2 Finding Multiple Solutions on Smooth Loss Landscape

The second key insight is corresponding to find multiple solutions on the same global loss landscape that are reside on the same low-loss level region. This can be done by leveraging multiple regression heads ($f_\theta$ and $h_\psi$ in Eq. 2). Our proposed framework for efficient ensembling is shown in Figure. 4. The representation $g_\phi(x)$ from the encoder is fed into each regression head to get $f_{\theta_i}(g_\phi(x))$ and $h_{\psi_i}(g_\phi(x))$ for $i$-th regression head. With these representations from $i$-th regression head, we can get $\tilde{y}_i$ in Eq. 2 for each head. Then, the loss function $L(\tilde{y}_i, y)$ is applied to each $\tilde{y}_i$. While each function, parameterized by $\{\phi, \theta_i, \psi_i\}$, serves a same purpose (estimating target $y$), they are represented differently, thus leading to enhanced expressive power. With this formation, we can get diverse parameter solution set $\{S_i\}$ where $S_i =$

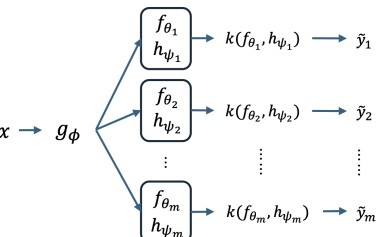

Figure 4: Overall proposed framework with $m$ regression heads with adaptive scaling mechanism. The operation $k(a, b)$ represents $\frac{a}{\sigma(b)+1}$.

$\{\phi, \theta_i, \psi_i\}$ on the smooth loss landscape induced by our adaptive scaling mechanism. Consequently, the proposed framework facilitates an efficient ensemble strategy to boost generalization ability of the model. Because each solution $S_i$ lies on a smooth and flat region of the loss landscape, which is known to be associated with improved stability and generalization ability of the model. So when the number of regression heads is $M$, the formulation of this framework and the loss function during training can be expressed as:

$$\tilde{y}_i = k(f_{\theta_i}(g_\phi(x)), h_{\psi_i}(g_\phi(x))) = \frac{f_{\theta_i}(g_\phi(x))}{\sigma(h_{\psi_i}(g_\phi(x))) + 1}, \qquad L = \frac{1}{M}\sum_{i=1}^M L_i, \qquad (3)$$

where $k(a, b)$ represents $\frac{a}{\sigma(b)+1}$ and $L_i = L(\tilde{y}_i, y)$ for $1 \leq i \leq M$. During inference, we averaged $\tilde{y}_i$ as follows:

$$\tilde{y}_{final} = \frac{1}{M}(\tilde{y}_1 + \tilde{y}_2 + \cdots + \tilde{y}_M). \quad (4)$$

Additionally, we intentionally initialized the parameters of regression heads differently to get diverse solution set. This is to prevent each head from converging to similar minimum in the global loss landscape. Also, it is worth noting that this ensemble strategy is valid when the adaptive scaling mechanism is combined. Since the loss landscape is highly complicated without adaptive scaling mechanism and potentially has multiple local minima with different loss level that cannot be distinguish from the perspective of the model, the differently initialized each regression head may converge to different local minima with different loss level, which could reduce the benefit of ensembling. We empirically validated this in Section 4.6.

## 4  Experiments

We experiment our method on benchmark datasets, Human3.6M (H36M) [14], MPI-INF-3DHP (3DHP) [24], 3D Poses in the Wild (3DPW) [30], and BEDLAM [2]. H36M [14] is broadly utilized for training and evaluation in 3D HPE task. They are consisted of over 3.6 million 3D pose and 2D pose pairs. We utilize the data from subjects 1, 5, 6, 7, and 8 as training set, while the data from subjects 9 and 11 are utilized as test set following the literature of 3D HPE. 3DHP [24] presents rather complex situations compared to H36M [14], including diverse indoor and outdoor scenarios. 3DPW [30] is a widely-used benchmark for 3D human pose estimation in unconstrained outdoor environments, featuring over 51,000 frames of video with accurate ground-truth 3D pose and shape. BEDLAM [2] is a large-scale synthetic video dataset for 3D human pose and shape estimation, featuring realistic clothing and high diversity in body shapes and motions. This dataset is specifically designed to train models that generalize to real-world scenario. We report metrics, Mean Per Joint Position Error (MPJPE) and Procrustes Aligned MPJPE (PA-MPJPE), on H36M [14] following convention. For 3DPW [24], we report Percentage of Correct Keypoints (PCK) within 150mm, Area Under the Curve (AUC), and MPJPE to evaluate the generalization capability of the model with our approach. For network architecture, we utilize representative architecture (*i.e.* MLP, CNN, GCN, and Transformer) of 3D HPE task to evaluate our proposed method.

### 4.1  Comparison on H36M

In this section, we apply our proposed method to a variety of baseline models [23, 27, 38, 37] in order to comprehensively evaluate the effectiveness and generalization capbility of our approach. Specifically, we integrate our method into multiple representative 3D HPE architectures and compare their performances before and after the application of our method in Table 1. To gain deeper insights into the individual contributions of each component within our framework, we conduct ablation studies under the following settings: (1) standard training without any modifications, (2) training with the adaptive scaling mechanism only, and (3) training with both the ensembling method and the adaptive scaling mechanism. The experimental results show that our proposed method consistently leads to performance improvements across a wide range of network architectures, highlighting its robustness and versatility. This consistent gain suggests that the integration of the ensembling strategy with the adaptive scaling mechanism generates a synergistic effect, effectively enhancing the generalization ability of the model. Moreover, these results indicate that our approach contributes to reshaping the loss landscape in a way that facilitates better optimization and improved final

Table 1: Before and after applying our method to various models. We report MPJPE and PA-MPJPE in a single frame setting for evaluation on H36M. Similar to other 3D HPE works [8, 27, 36, 21], we use cascaded pyramid network [3] as a 2D pose detector.

| Architecture | Method | MPJPE | | | PA-MPJPE | | |
|---|---|---|---|---|---|---|---|
| | | Basic | +Scale | +Both | Basic | +Scale | +Both |
| MLP | BaselineNet [23] | 55.4 | 54.2 | **53.6** | 44.2 | 43.1 | **42.7** |
| CNN | VPose [27] | 55.7 | 54.9 | **54.4** | 44.0 | 43.4 | **43.3** |
| GCN | SemGCN [37] | 64.8 | 61.6 | **59.9** | 51.2 | 48.9 | **48.3** |
| Transformer | PoseFormer [38] | 56.2 | 55.5 | **54.5** | 44.3 | 44.4 | **43.9** |

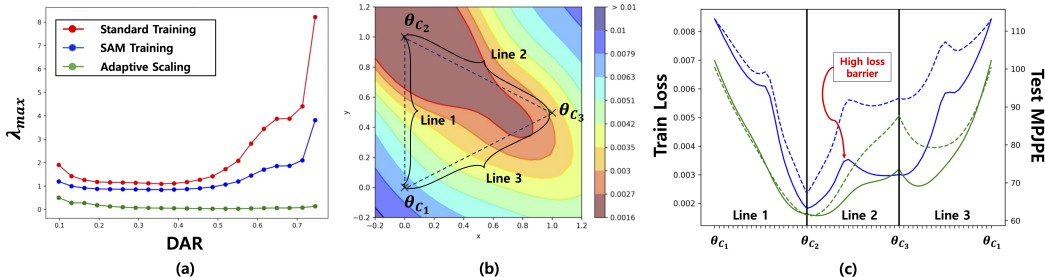

Figure 5: (a) Top-1 eigenvalue of each local loss landscape when $K$ is 21 after applying SAM [5] and adaptive scaling mechanism. (b) Global loss landscape of after applying adaptive scaling mechanism alone. (c) Training loss and test accuracy along the line between local minimum ($\theta_{C_1}$, $\theta_{C_2}$, and $\theta_{C_3}$) of local loss landscape. The dashed lines refer to the test accuracy. The blue line is for standard model and green line is for model with adaptive scaling mechanism. Note that these results are from BaselineNet [23].

accuracy. Overall, the results validate the practical utility of our method as a simple yet meaningful enhancement to existing 3D HPE frameworks.

## 4.2 Cross-Dataset Evaluation on 3DPW

To further assess how effectively our proposed method enhances the generalization ability of 3D HPE models, we conducted a cross-dataset evaluation. This type of evaluation is crucial for understanding whether a model trained on one dataset can perform well on data from a different domain, which is often more reflective of real-world deployment scenarios. We trained each model with a combined ground-truth training set of the H36M, 3DHP, and BEDLAM and then, evaluated its performance on the 3DPW dataset. During testing, we used ground-truth 2D poses from 3DPW as input and predicted the corresponding 3D poses using the trained models. We report three standard metrics

Table 2: Cross-dataset evaluation on 3DPW.

| Method | PCK | AUC | MPJPE |
|---|---|---|---|
| BaselineNet [23] | 85.8 | 47.6 | 87.2 |
| +Ours | **87.1** | **50.8** | **81.9** |
| VPose [27] | 85.3 | 47.4 | 88.8 |
| +Ours | **86.5** | **47.9** | **86.0** |
| SemGCN [37] | 78.2 | 41.4 | 104.3 |
| +Ours | **84.4** | **48.7** | **88.7** |
| PoseFormer [38] | 84.3 | 47.2 | 90.9 |
| +Ours | **86.0** | **50.3** | **84.1** |

to comprehensively assess performance: PCK, AUC, and MPJPE. As presented in Table 2, our method consistently leads to significant improvements in all three metrics across various baseline architectures. These results strongly suggest that our approach enhances the generalization ability of the model beyond the training data distribution, making it more robust to unseen domains and varying data characteristics. Such improvements in generalization capability are particularly valuable for practical applications where the test-time data may differ from the training distribution.

## 4.3 Flatness of Local minima

To investigate whether our adaptive scaling mechanism effectively smooths the loss landscape, we estimated the top-1 eigenvalue of the Hessian matrix when only the adaptive scaling mechanism was applied. As shown in Figure 5(a), the curvature of each local loss landscape became noticeably flatter compared to that of baseline. Additionally, we visualized the global loss landscape under the adaptive scaling mechanism alone in case of BaselineNet [23] to compare the global loss landscape without our adaptive scaling mechanism in Figure 1(d). As illustrated in Figure 5(b), the landscape exhibited a smoother topology with a single local minimum. This provides a significant advantage in network convergence and, as demonstrated in Section 4.2, the flatness of the loss landscape correlates with improved generalization performance. Furthermore, in Figure 5(c), we investigate the training loss and test MPJPE along the line between each local minimum. Our adaptive scaling mechanism exhibits better training loss and test MPJPE through the all points on the line between each local minimum. Also there was no high loss barrier between each local minimum. This validates our intuition in Section 2.4.

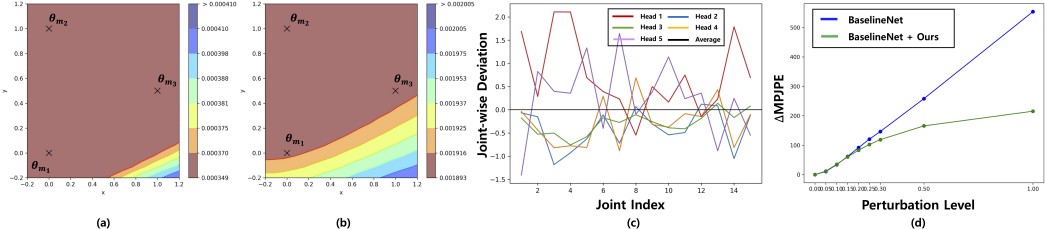

Figure 6: (a) Diverse solution set on the train loss landscape under our adaptive scaling mechanism. $\theta_{m_1}$, $\theta_{m_2}$, and $\theta_{m_3}$ are solutions when the number of heads $M$ is 3. (b) Diverse solution set on the test loss landscape under our adaptive scaling mechanism. (c) Joint-wise deviation for predictions of PoseFormer [38]. (d) Robustness against input Gaussian noise. $\Delta$MPJPE represents the gap between MPJPE for perturbed input and original MPJPE.

## 4.4 Example of Diverse Solutions on Loss Landscape

DNNs typically consist of millions to even billions of parameters, making it challenging to analyze their loss landscapes directly. Nevertheless, several effective techniques have been proposed to visualize the loss landscape in a interpretable manner [20, 6]. Similar to the [6], we visualize the loss landscape of BaselineNet [23] with 3 heads in Figure 6. Because visualizing the loss landscape with all solutions becomes challenge when the number of solutions exceeds 3. The visualization results show that all solutions reside on low-loss and flat regions of the training loss landscape. Similarly, we observe that the corresponding test loss landscapes also place the solution set in flat regions with low-loss level. We further investigate the joint-wise prediction diversity for PoseFormer [38] in Figure 6(c). The predictions exhibited diversity across different heads. These results suggest that ensembling the solutions can lead to a highly stable model, benefiting from low bias and reduced variance. Such stable models are crucial in real-world applications, where consistent and reliable predictions are essential under varying and potentially noisy conditions.

## 4.5 Comparison with Other Loss Landscape Flattening Method

There are various approaches to promoting flatness in the loss landscape. A representative example is Sharpness-Aware Minimization (SAM [5]), which explicitly encourages flatness by applying perturbations and penalizing sharp regions in the loss landscape. However, such methods operate without considering the characteristics of individual local loss landscapes, instead applying a global flattening effect. As a result, they may disproportionately flatten regions around specific local minimum, potentially overlooking others. In fact, we investigate

Table 3: Comparison with SAM on H36M.

| Method | MPJPE | PA-MPJPE |
|---|---|---|
| BaselineNet [23] | 54.8 | 43.8 |
| +SAM [5] | 56.1 | 44.4 |
| +Scale only | **54.2** | **43.1** |
| VPose [27] | 55.7 | 44.0 |
| +SAM [5] | 57.3 | 45.7 |
| +Scale only | **54.9** | **43.4** |

the top-1 eigenvalue of Hessian matrix when applying SAM in Figure 5(a). As we expected, the SAM tends to flatten the loss landscape in a biased manner, favoring certain local regions over others. We also experiment the performance of 3D HPE + SAM in Table 3. The results show that applying SAM to a 3D HPE model yielded no performance gain, which confirmed our hypothesis. Note that the models are trained for a longer duration because of the slow convergence of SAM and the perturbation radius is set as 0.05 for SAM training.

## 4.6 Ensembling without Adaptive Scaling Mechanism

We further validate that the proposed ensembling method performs effectively when combined with the adaptive scaling mechanism in Table 4. As we expected, the ensembling approach alone does not achieve satisfactory results in the absence of adaptive scaling, primarily due to the highly complex and irregular nature of the global loss landscape. This complexity can cause each regression head to converge to different local minima, leading to a severe gradient conflicts between heads or a lack of coherence among the predictions and diminishing the overall benefit of ensembling. To further clarify this, we evaluate the combination of another loss landscape flattening method, SAM,

with the ensemble strategy. The results indicate that 3D HPE + SAM + Ensemble achieves better performance than ensemble alone, which supports that the loss landscape of 3D HPE is highly complicated. All these findings underscore the crucial role of adaptive scaling mechanism in aligning the optimization trajectories of individual heads and enhancing ensemble performance. The adaptive scaling mechanism harmonizes the learning dynamics across heads, facilitating more coordinated updates and yielding a more effective ensemble representation.

Table 4: Results of esembling without adaptive scaling mechanism on H36M.

| Method | MPJPE | PA-MPJPE |
|---|---|---|
| BaselineNet [23] | 54.8 | 43.8 |
| +SAM [5] and Ens. | 54.5 | 43.2 |
| +Ens. only | 73.3 | 58.1 |
| VPose [27] | 55.7 | 44.0 |
| +SAM [5] and Ens. | 55.3 | 44.0 |
| +Ens. only | 69.8 | 55.2 |

### 4.7 Robustness of Ensembling Strategy

We further experiment the robustness of our model to the input noise. Lifting-based 3D HPE, a prevalent approach in the literature, typically relies on the output of a 2D pose detector as input. As a result, the input inevitably contains noise. Therefore, evaluating the robustness of the 3D HPE network to input noise is crucial. To simulate noisy inputs, we perturb the 2D pose inputs with Gaussian noise of varying magnitudes in input scale-invariant manner. In this reason, we experiment our method to evaluate the robustness to input noise on the dataset 3DHP [24] for the BaselienNet [23] in Figure 6(d). Note that the model is trained with H36M [14]. The model with our method shows better generalization capability. Especially, our model is robust when the noise level is high, which suggests that our method can generalize better for unseen poses. Stability under high noise levels is crucial, as it enables fail-safe mechanisms, in which the robot system can still recognize a human shape in a danger zone even when the predicted pose is inaccurate.

### 4.8 Converging Speed

As mentioned above, the adaptive scaling mechanism serves to smooth the loss landscape. Consequently, it is natural to think that the optimization of models with adaptive scaling mechanism becomes more easier than before. To illustrate this, we compare the training loss trajectories of the model with and without adaptive scaling mechanism during 20 epochs on H36M [14] in Figure 7. As we expected, the model with adaptive scaling mechanism shows a faster convergence. The training loss with adaptive scaling mechanism shows lower training loss throughout the epochs. Especially, the test accuracy is significantly lower at the beginning of the training. With this property, while our adaptive scaling mechanism facilitates faster convergence, the model with adaptive scaling mechanism may need to be early-stopped to avoid undesirable overfitting.

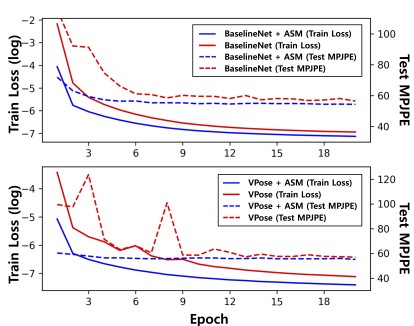

Figure 7: Loss trajectories with adaptive scaling mechanism. ASM refers to the adaptive scaling mechanism.

## 5 Related Work

### 5.1 3D Human Pose Estimation

A wide range of methods have been proposed for 3D HPE. The prevalent research direction in this area is to lift 2D keypoints in image coordinate system to 3D pose coordinates in camera coordinate system [23, 27, 37, 38]. There are various representative network architectures starting from transformer-based model [22, 38], GCN-based model [37], CNN-based model [27], and MLP-based model [23, 35]. These researches proposed elastic network design to learn better posture information to enhance the generalization capability of the 3D HPE model. On the other hand, augmentation-based approaches [7, 16, 21] have been proposed to address the limited availability of training data in 3D HPE task. These methods primarily aim to improve model generalization by enhancing data diversity through various augmentation techniques. While these methods have demonstrated notable performance improvements, they overlooked the perspective of the loss landscape, which is closely

related to generalization. In this work, we aim to provide a novel perspective on 3D HPE by analyzing its loss landscape and propose a simple yet effective approach based on this analysis.

## 5.2 Loss Landscape and Generalization

Improving generalization through loss landscape smoothing has become a prominent research direction in deep learning, as flatter minima are often associated with improved robustness and better out-of-distribution performance. This phenomenon was revealed by a pioneer work [9]. After this, a number of influential techniques have been proposed to address this, most notably Sharpness-Aware Minimization (SAM) [5], which explicitly penalizes sharp minima by optimizing for the worst-case loss within a neighborhood around each parameter update. With SAM, various SAM-variant method was proposed in recent years [17, 32, 25]. However, these methods incur high computational costs due to the use of double backpropagation during training. Nevertheless, we argue that the insight they provide is valuable, and applying it to the 3D HPE task is worthwhile despite the overhead. In this work, we aim to analyze the loss landscape of 3D HPE to leverage this insight and propose a simple yet effective method based on the analysis.

## 5.3 Deep Ensemble Learning

Deep ensemble learning has emerged as a widely used technique to improve the robustness and generalization of neural networks, particularly in settings where model uncertainty and out-of-distribution (OOD) performance are critical. [18] introduced a simple yet effective deep ensemble method by independently training multiple neural networks with random initialization, showing improved predictive uncertainty estimation and better calibration compared to single models. Based on this, [11] introduced a simple yet effective deep ensemble method by leveraging multiple checkpoints during training process. [6] proposed fast geometric ensembling method by utilizing connected modes property to enhance the generalization capability. [15] proposed weight ensembling method to find more flatter region on the loss landscape. Despite their effectiveness, these approaches tend to be computationally expensive or rely on two-stage training pipelines, which limit their practicality. In this work, we aim to propose parameter-efficient ensembling method to boost generalization capability of the 3D HPE model.

## 6 Conclusion

In this work, we analyzed the loss landscape of the 3D HPE task. Our empirical investigation showed that the loss landscape of the 3D HPE can be highly complicated and potentially have multiple local minima, which poses significant challenge during training process. This observations highlight the importance of considering loss landscape geometric properties during optimization. To mitigate these issues, we introduced a novel and simple adaptive scaling mechanism to flatten the loss landscapes. With this adaptive scaling approach, we also provide a efficient ensembling method to boost the generalization ability of the model. The proposed method effectively addresses the challenges identified in our analysis, and our experimental results validate the soundness and practicality of our approach. Furthermore, our adaptive scaling mechanism could be beneficial for tasks with highly complicated loss landscapes, such as 3D HPE, although it must also account for task-specific characteristics, including the location where adaptive scaling mechanisms are applied.

## Acknowledgements

This work was supported by Institute of Information & communications Technology Planning & Evaluation (IITP) grant funded by the Korea government(MSIT) (No. RS-2019-II190079, Artificial Intelligence Graduate School Program (Korea University), No. IITP-2025-RS-2025-02304828, Artificial Intelligence Star Fellowship support program to nurture the best talents, No. IITP-2025-RS-2024-00436857, Information Technology Research Center, and No. RS-2024-00457882, AI Research Hub Project).

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
