# OpenReview forum: "Towards Generalizable 3D Human Pose Estimation via Ensembles on Flat Loss Landscapes"
_NeurIPS.cc/2025/Conference — NeurIPS 2025 poster_

### Official Review · Reviewer_sYww · 2025-06-11

**Clarity:** 2
**Significance:** 2
**Originality:** 2
**Rating:** 4
**Confidence:** 2

**Summary:**

Overall, this submission focuses on improving the generalization performance of 3D human pose estimation models. It achieves this with the help of flat loss landscape and ensemble.

**Questions:**

Overall, the reviewer currently votes for borderline rejection for this submission due to the following concerns.

1. Firstly, the reviewer is confused about the relationship between the proposed method and investigation and the 3D HPE task. Specifically, while it can be admitted that generalization is important in 3D HPE, it is also important for almost any other computer vision tasks. Thus, why the authors focus on improving the generalization performance of models while specifically focuses on 3D HPE is confusing, especially considering that the reviewer seems to also fail to get the relationshipf between the proposed method and the 3D HPE task. Instead, it seems that both parts of the methods are quite general methods that are not specifically relevant to the 3D HPE task.

2. Moreover, the reviewer also appreciate the authors to better elaborate and explain on the significance behind its key insights. Specifically, the first key insight that the authors seem to aim to elaborate in Sec 2.4 seems to be existing and relatively general. The reviewer thus fails to understand why this insight need to be re-mentioned particularly in the 3D HPE context. Similarly, the second key insight that ensembling can improve generalization capability looks to be also very very general and common sense from the reviewer's perspective.

3. Meanwhile, the authors claim that "our method requires nearly the same amount of time as training a single model". Yet, this claim seems to be very inadquately supported. Specifically, the reviewer seems to only see from Table 2 in Appendix that, across different baselines, the proposed method all obviously increase the inference time. Yet, the reviewer seems to see no clear comparison w.r.t. the training time. The reviewer appreciates explanation over the improper support of this claim.

4. Lastly, in the introduction line 23-28, the authors seem to want to show that many 3D HPE methods have focused on enhancing robustness. While this is the case, in the experimental section, the authors seem to compare to none of these robustness enhancing 3D HPE works. Instead, it only uses several methods as baselines and apply the proposed method on these baselines. This leads the experiments to be significantly non-comprehensive.

Given the reviewer's above concerns, the reviewer currently tends to reject this submission.

**Ethical Concerns:**

["NO or VERY MINOR ethics concerns only"]

**Final Justification:**

After reading the authors rebuttal, the reviewer now gets the relationship between their method and this particular 3D HPE task. The reviewer has raised point from 3 to 4. Yet, the reviewer strongly suggests the authors to carefully edit their camera ready based on the answers provided in the rebuttal period.

**Limitations:**

Yes.

**Paper Formatting Concerns:**

N.A.

**Quality:**

2

**Strengths And Weaknesses:**

Strength:

Overall, this submission is easy to follow while it emphrically also demonstrates good performance.

Weaknesses:

(See questions section for more details).

---

> ### Author Rebuttal · Authors · 2025-07-31
>
> We appreciate the reviewer for taking the time to carefully review our paper. We deeply appreciate the constructive comments. We are pleased to hear that our paper is easy to follow.
>
> > # W1. The relationship between our method and the rationales of our analysis and 3D HPE task.
>
> A key distinction between 3D HPE and other computer vision tasks lies in its inherent ambiguity, which means that a single 2D input can correspond to multiple valid 3D poses (the model can have multiple valid loss values for a single input). This one-to-many mapping sets 3D HPE apart from problems with more deterministic input-output relationships. We quantify this ambiguity using the Depth Ambiguity Ratio (DAR), which measures the degree of depth uncertainty in the data. The higher DAR, the higher variance of possible loss values for a single input. This justify our analysis based on the DAR value, which directly affects the geometry of loss landscape. Our analysis based on the DAR shows that loss landscape of 3D HPE is highly complex, which is underexplored previously. Such loss landscape poses significant challenges for optimization in 3D HPE task, as standard training methods struggle to find flat and generalizable minima in the presence of this complexity. To address this, our method adapts adaptive scaling mechanism to smooth loss landscape, improving generalization ability specifically in 3D HPE task.
>
>
> > # W2. The connection between our key insights and 3D HPE task.
>
> Our first key insight focuses on “Flatness around local minimum of each **local** loss landscape”, which is different from “Flatness around local minimum of **global** loss landscape” in the standard deep learning literature.
> The key difference between these two lies on the fact that our first key insight considers the different local loss landscape geometries, which is not considered in the other. Especially, this consideration has significant impact on 3D HPE task (unlike other computer vision tasks) considering the impact of DAR on the global loss landscape of 3D HPE task as we mentioned above.
> As a result, previous loss landscape flattening method (e.g. SAM) smoothed the loss landscape in a biased manner as shown in Fig. 5(a) in the paper, whereas our adaptive scaling mechanism preserves balanced flatness across local loss landscape. This is desirable property for one-to-many problem such as 3D HPE.
>
> For our second key insight, “Finding multiple solutions”, it is not about just “doing ensemble”.
> As mentioned in the paper, the second key insight is motivated from the adaptive scaling mechasim and **plays a key role** to connect between adaptive scaling mechanism and ensembling on the smoothed loss landscape.
> In fact, our ensembling strategy **without** adaptive scaling mechanism doesn’t work as shown in Tab. 4 in the paper.
> This means that our ensembling strategy from second key insight is mutually complementary to adaptive scaling mechanism and facilitates an effective ensemble to further boost generalization ability of the 3D HPE model with adaptive scaling mechanism.
>
> As a result, our key insights are derived from the 3D HPE-relevant issues and motivations.
>
> we will clarify this in the future version.
>
> > # W3. Training time.
>
> We present training time comparison for one epoch with various model architectures.
> We use a single NVIDIA A6000 GPU for this comprarison.
> For FLOPs, please refer to the supplementary material Sec. E.
>
> |Model|Time (sec.)|Model|Time (sec.)|Model|Time (sec.)|Model|Time (sec.)|
> |-|:-:|-|:-:|-|:-:|-|:-:|
> |Baseline|18|VPose|22|SemGCN|39|PoseFormer|40|
> |Baseline+Ours|19|VPose+Ours|24|SemGCN+Ours|42|PoseFormer+Ours|43|
>
>
> > # W4. Comparison with other works.
>
> The goal of our work is fundamentally different from prior methods aimed at improving robustness in 3D HPE. While other methods focus on enhancing robustness through data augmentation (e.g. PoseAug[2]), proposing their own model architecture (e.g. RIE[1]), and multi-hypothesis concept (e.g. ProPose[3]), we approach the problem from a loss landscape perspective—by addressing the optimization difficulty caused by loss landscape complexity.
> Importantly, our method is orthogonal to these prior approaches and can be combined with them. We present this by applying our method on top of RIE, PoseAug, and ProPose, resulting in consistent improvements across benchmarks.
> The models are trained on training dataset of H36M and evaluated on test sets of H36M or 3DHP. For Ours, we set the number of heads as 3.
>
> |Model|MPJPE(H36M)&darr;|PA-MPJPE(H36M)&darr;|MPJPE(3DHP)&darr;|PA-MPJPE(3DHP)&darr;|
> |-|:-:|:-:|:-:|:-:|
> |RIE[1]|53.1|42.7|89.4|68.3|
> |RIE[1]+Ours|**52.9**|**41.8**|**86.8**|**67.3**|
> |BaselineNet+PoseAug[2]|54.1|42.8|82.8|63.1|
> |BaselineNet+PoseAug[2]+Ours|**53.9**|**42.6**|**81.1**|**61.8**|
> |VPose+PoseAug[2]|54.7|43.1|83.8|63.8|
> |VPose+PoseAug[2]+Ours|**54.2**|**42.8**|**81.7**|**60.7**|
>
> For multi-hypothesis concept, we report results in single-hypothesis setting below.
>
> |Model|PCK(3DHP)&uarr;|AUC(3DHP)&uarr;|
> |-|:-:|:-:|
> |ProPose[3]|84.4|52.1|
> |ProPose[3]+Ours|**84.6**|**52.2**|
>
> [1] W. Shan et al., “Improving robustness and accuracy via relative information encoding in 3D human pose estimation,” ACM, 2021.
>
> [2] K. Gong et al., “Poseaug: A differentiable pose augmentation framework for 3D human pose estimation,” CVPR, 2021.
>
> [3] J. Han et al., “Propose: Probabilistic 3D human pose estimation with instance-level distribution and normalizing flow,” AAAI, 2025.

---

> > ### Comment · Reviewer_sYww · 2025-08-07
> >
> > Hi authors,
> >
> > I am sorry that previously, I thought you can see the final justification and I thus left my feedback to your rebuttal there.
> >
> > As mentioned there, after reading the authors rebuttal, the reviewer now gets the relationship between their method and this particular 3D HPE task. The reviewer has raised point from 3 to 4. Yet, the reviewer strongly suggests the authors to carefully edit their camera ready based on the answers provided in the rebuttal period.

---

> > > ### Author Response · Authors · 2025-08-08
> > > **Thank you for your thoughtful discussion.**
> > >
> > > We sincerely thank you for your careful consideration and constructive feedback. We are grateful for your increased evaluation and for recognizing the relevance of our method to the 3D HPE task. We will update the manuscript by incorporating the points raised during the rebuttal period. Thank you again for your valuable suggestions and time.

---

### Official Review · Reviewer_X5JK · 2025-06-17

**Clarity:** 3
**Significance:** 3
**Originality:** 3
**Rating:** 4
**Confidence:** 3

**Summary:**

This paper explores the loss landscape in 3D human pose estimation task, and proposes an adaptive scaling mechanism to smooth the loss landscape. The authors visualize the loss landscape and show its impact on performance improvement. Based on this, they introduce a simple, model-agnostic ensemble method that leverages the smoothed landscape to combine diverse solutions. The approach improves performance across various HPE methods on benchmarks like Human3.6M and MPI-INF-3DHP.

**Questions:**

Please see the weaknesses listed above. I would be inclined to increase my rating if these concerns are addressed.

**Ethical Concerns:**

["NO or VERY MINOR ethics concerns only"]

**Final Justification:**

The authors’ response addressed my concerns, and I have decided to raise my ranking from 3 to 4.

**Limitations:**

Yes

**Paper Formatting Concerns:**

No major formatting issues noticed.

**Quality:**

3

**Strengths And Weaknesses:**

Strengths
1. The paper is well-structured and easy to follow.
2. The adaptive scaling mechanism and ensemble method are simple enough to reproduce.
3. The experiments clearly demonstrate the smoothed loss landscape and the performance improvement.

Weaknesses
1. As mentioned in the limitations, the paper lacks theoretical analysis of the proposed method and its results, making it difficult to understand the underlying principles and justify its effectiveness.
2. Section 3.2 does not explain how the different regression heads are obtained. Are they trained on different datasets, or are they simply initialized with different random seeds?
3. In Figure 6(d), the performance of both methods is similar under small perturbations. At a perturbation level of 0.2, ΔMPJPE exceeds 100, which is significant. Although some robustness is shown under large perturbations, such a large error renders the results practically unusable.
4. It is unclear how the value of K is selected and how it impacts the results. The paper lacks experiments or analysis to support this design choice.

---

> ### Author Rebuttal · Authors · 2025-07-31
>
> We appreciate the reviewer for taking the time to carefully review our paper. We deeply appreciate the constructive comments. We are pleased to hear that our contributions were well received. In this response, we aim to carefully address the feedback provided by the reviewer.
>
> > # W1. A simple theoretical understanding of our method.
>
> We provide a simple theoretical understanding of our method below.
>
> - **For Adaptive Scaling Mechanism**,
>
>
> The formulation of Eq. 3 in the main paper is $\frac{f_{\theta_i}(X)}{\sigma(h_{\psi_i}(X))+1}$, where $\sigma$ is ReLU function, $g_\phi(x) = X$, and $x$ is input.
>
> And let’s say that $\frac{f_{\theta_i}(X)}{\sigma(h_{\psi_i}(X))+1} = \frac{f_{\theta_j}(X)}{\sigma(h_{\psi_j}(X))+1} = \tilde{y}$. So each head has same output and same loss value.
>
> When $\sigma(h_{\psi_i}(X))+1 = a$ and  $\sigma(h_{\psi_j}(X))+1 = a+\Delta a$, then $f_{\theta_i}(X) = a\tilde{y}$ and $f_{\theta_j}(X) = a\tilde{y} + \Delta a\tilde{y}$. ($a$ is positive)
>
> By substitution, $\sigma(h_{\psi_j}(X)) - \sigma(h_{\psi_i}(X)) = \Delta a$ and $f_{\theta_j}(X) - f_{\theta_i}(X) = \Delta a\tilde{y}$.
>
> By the Taylor expansion, $f_{\theta_j}(X) \approx  f_{\theta_i}(X) + \nabla_\theta f_{\theta_i}(X)^T (\theta_j - \theta_i)$.
> Similary, $\sigma(h_{\psi_j}(X)) \approx \sigma(h_{\psi_i}(X)) + \nabla_\psi \sigma(h_{\psi_i}(X))^T (\psi_j - \psi_i)$.
>
> For simplicity, we address the case of $\theta$ here.
>
> By the substitution, $\nabla_\theta f_{\theta_i}(X)^T (\theta_j - \theta_i) \approx \Delta a\tilde{y}$.
>
> Here, the gradient $\nabla_\theta f_{\theta_i}(X)$ has large enough rank and is non-zero in practical.
>
> So when $\Delta a \rightarrow 0$, the $(\theta_j - \theta_i) \rightarrow 0$. This implies that there exists connected region in $\theta$ space where shares same loss value depending on the value of $\Delta a$.
>
>
>
> - **For Ensemble strategy**,
>
>
> Let’s say that $S_j = S_i + \Delta S_{ij}$, where $S_i = (\phi, \theta_i, \psi_i)$.
>
> By Taylor expansion of  $F_{S_j}(x) = y,   F_{S_i+\Delta S_{ij}} \approx F_{S_i}(x) + \nabla_S F_{S_i}(x)^T \Delta S_{ij}$.
>
> If $\nabla_S F_{S_i}(x)^T$ and $\Delta S_{ij}$ are non-zero, then $F_{S_i}(x) \neq F_{S_j}(x)$, which means they are different function.
> In practical, the gradient is not zero vector.
> And to make $\Delta S_{ij} \neq 0$, we intentionally initialized each head differently as mentioned in the paper.
>
> As a result, each $F_{S_i}(x)$ is different functions so this can justify our ensemble strategy.
>
> > # W2. How to obtain the different regression heads?
>
> As mentioned in the main paper and Supplementary Sec. D, we obtain different regression heads by initializing each head differently. Specifically, we use orthogonal initialization [1] for each head to promote diversity, and further add scaled noise sampled from $N(0,I)$ to encourage variation in weight magnitudes.
>
> All heads are trained on the same dataset so they observe the same data and labels. We also experimented with training each head on a different data split, but we observed no significant improvement in diversity or downstream performance. Therefore, we chose the simpler setup of using the same dataset with diversified initialization.
>
> We will clarify this point more explicitly in the revised version.
>
> [1] A. Saxe et al., “Exact solutions to the nonlinear dynamics of learning in deep linear neural networks,” ICLR, 2014.
>
> > # W3. Noise robustness of our method.
>
> We present detailed results at small perturbation level for BaselineNet in MPJPE.
>
> |Model|0|0.01|0.03|0.05|0.07|0.09|0.1|
> |-|-|-|-|-|-|-|-|
> |BaselineNet|89.8|90.2|93.6|100.4|108.6|118.3|123.7|
> |BaselineNet+Ours|**85.6**|**85.7**|**89.9**|**97.9**|**106.4**|**116.2**|**121.5**|
>
> It is true that while our method consistently outperforms the baseline across all noise levels in terms of MPJPE, the relative robustness is not superior at low perturbation levels.
> However, we would like to clarify that our goal was to demonstrate the overall stability of our method under noise, particularly in scenarios that simulate substantial input variation. In fact, we interpret robustness under large perturbations as a proxy for the model's ability to generalize to unseen or out-of-distribution poses as also described in the paper (line319-320), which are common in real-world applications such as outdoor or sports environments. This was the motivation behind the analysis.
>
> We will clarify this in the future version.
>
> > # W4. The impact of K.
>
> In Section 2 of the main paper, we use K = 3 for visualization purposes only—this allows us to clearly and easily depict the geometry of local loss landscapes and their corresponding local minima within a shared coordinate system. To address how different values of K impact the global loss landscape, we conducted further experiments with K = 4, 5, 6, 7, 8. We present loss values on the global loss landscape at linearly interpolated parameter between local minima of each local loss landscape. The results are presented in the table below. Note that $L_{max}(\theta_{C_i}, \theta_{C_j})$ refers to $max_{\alpha \in [0,1]}\left(L((1-\alpha)\theta_{C_i}+\alpha\theta_{C_j})\right)$. As we are not allowed to present figure, please understand the way we present results.
>
> When K=4.
> |Model|$\theta_{C_1}$|$L_{max}(\theta_{C_1}, \theta_{C_2})$|$\theta_{C_2}$|$L_{max}(\theta_{C_2}, \theta_{C_3})$|$\theta_{C_3}$|$L_{max}(\theta_{C_3}, \theta_{C_4})$|$\theta_{C_4}$|$L_{max}(\theta_{C_4}, \theta_{C_1})$|$\theta_{C_1}$|
> |-|:-:|:-:|:-:|:-:|:-:|:-:|:-:|:-:|:-:|
> |BaselineNet (Loss)|0.015|**0.056**|0.007|**0.060**|0.005|**0.038**|0.010|**0.086**|0.015|
> |VPose (Loss)|0.012|**0.012**|0.003|**0.007**|0.002|**0.008**|0.005|**0.012**|0.012|
>
> When K=5.
> |Model|$\theta_{C_1}$|$L_{max}(\theta_{C_1}, \theta_{C_2})$|$\theta_{C_2}$|$L_{max}(\theta_{C_2}, \theta_{C_3})$|$\theta_{C_3}$|$L_{max}(\theta_{C_3}, \theta_{C_4})$|$\theta_{C_4}$|$L_{max}(\theta_{C_4}, \theta_{C_5})$|$\theta_{C_5}$|$L_{max}(\theta_{C_5}, \theta_{C_1})$|$\theta_{C_1}$|
> |-|:-:|:-:|:-:|:-:|:-:|:-:|:-:|:-:|:-:|:-:|:-:|
> |BaselineNet (Loss)|0.018|**0.019**|0.010|**0.041**|0.007|**0.041**|0.006|**0.033**|0.013|**0.114**|0.018|
> |VPose (Loss)|0.012|**0.012**|0.005|**0.007**|0.002|**0.030**|0.002|**0.012**|0.006|**0.013**|0.012|
>
> When K=6.
> |Model|$\theta_{C_1}$|$L_{max}(\theta_{C_1}, \theta_{C_2})$|$\theta_{C_2}$|$L_{max}(\theta_{C_2}, \theta_{C_3})$|$\theta_{C_3}$|$L_{max}(\theta_{C_3}, \theta_{C_4})$|$\theta_{C_4}$|$L_{max}(\theta_{C_4}, \theta_{C_5})$|$\theta_{C_5}$|$L_{max}(\theta_{C_5}, \theta_{C_6})$|$\theta_{C_6}$|$L_{max}(\theta_{C_6}, \theta_{C_1})$|$\theta_{C_1}$|
> |-|:-:|:-:|:-:|:-:|:-:|:-:|:-:|:-:|:-:|:-:|:-:|:-:|:-:|
> |BaselineNet (Loss)|0.019|**0.066**|0.127|**0.519**|0.008|**0.036**|0.007|**0.030**|0.007|**0.036**|0.019|**0.127**|0.019|
> |VPose (Loss)|0.014|**0.015**|0.006|**0.006**|0.003|**0.005**|0.002|**0.004**|0.003|**0.008**|0.008|**0.014**|0.014|
>
> When K=7.
> |Model|$\theta_{C_1}$|$L_{max}(\theta_{C_1}, \theta_{C_2})$|$\theta_{C_2}$|$L_{max}(\theta_{C_2}, \theta_{C_3})$|$\theta_{C_3}$|$L_{max}(\theta_{C_3}, \theta_{C_4})$|$\theta_{C_4}$|$L_{max}(\theta_{C_4}, \theta_{C_5})$|$\theta_{C_5}$|$L_{max}(\theta_{C_5}, \theta_{C_6})$|$\theta_{C_6}$|$L_{max}(\theta_{C_6}, \theta_{C_7})$|$\theta_{C_7}$|$L_{max}(\theta_{C_7}, \theta_{C_1})$|$\theta_{C_1}$|
> |-|:-:|:-:|:-:|:-:|:-:|:-:|:-:|:-:|:-:|:-:|:-:|:-:|:-:|:-:|:-:|
> |BaselineNet (Loss)|0.022|**0.062**|0.016|**0.018**|0.010|**0.047**|0.008|**0.053**|0.007|**0.030**|0.009|**0.036**|0.019|**0.145**|0.022|
> |VPose (Loss)|0.014|**0.014**|0.007|**0.009**|0.003|**0.004**|0.002|**0.008**|0.002|**0.005**|0.003|**0.008**|0.008|**0.014**|0.014|
>
> When K=8.
> |Model|$\theta_{C_1}$|$L_{max}(\theta_{C_1}, \theta_{C_2})$|$\theta_{C_2}$|$L_{max}(\theta_{C_2}, \theta_{C_3})$|$\theta_{C_3}$|$L_{max}(\theta_{C_3}, \theta_{C_4})$|$\theta_{C_4}$|$L_{max}(\theta_{C_4}, \theta_{C_5})$|$\theta_{C_5}$|$L_{max}(\theta_{C_5}, \theta_{C_6})$|$\theta_{C_6}$|$L_{max}(\theta_{C_6}, \theta_{C_7})$|$\theta_{C_7}$|$L_{max}(\theta_{C_7}, \theta_{C_8})$|$\theta_{C_8}$|$L_{max}(\theta_{C_8}, \theta_{C_1})$|$\theta_{C_1}$|
> |-|:-:|:-:|:-:|:-:|:-:|:-:|:-:|:-:|:-:|:-:|:-:|:-:|:-:|:-:|:-:|:-:|:-:|
> |BaselineNet (Loss)|0.023|**0.060**|0.016|**0.054**|0.011|**0.087**|0.008|**0.042**|0.007|**0.036**|0.009|**0.040**|0.009|**0.053**|0.023|**0.125**|0.023|
> |VPose (Loss)|0.015|**0.017**|0.008|**0.011**|0.005|**0.006**|0.003|**0.005**|0.002|**0.007**|0.002|**0.004**|0.003|**0.010**|0.010|**0.015**|0.015|
>
> These values consistently demonstrate that the local minima of each local loss landscape remain disconnected on the global loss landscape, confirming our first key insight, the "Flatness around each local minimum of local loss landscape", holds irrespective of K. Furthermore, considering the inherent redundancy and imblance in 3D HPE datasets, this behavior is likely to generalize across different values of K. Since the maximum meaningful value of K is the dataset size, we design our method under the assumption that K equals to the dataset size.
>
> We will clarify this in the future version.

---

> > ### Comment · Reviewer_X5JK · 2025-08-05
> >
> > We thank the authors for the theoretical explanation and the additional experiments, which have addressed some of my concerns. However, I still have some doubts regarding noise robustness:
> > 1. In Fig. 6 (d), although your model appears more stable at high noise levels, the $\Delta$MPJPE exceeds 100, which indicates a poor result. While the baseline is indeed worse, both results are essentially unusable under such noise levels.
> > 2. In Fig. 6 (d), when the perturbation level is 0.1, the $\Delta$MPJPE clearly exceeds 50, yet the value reported in the accompanying table is less than 40. Could you clarify this inconsistency?

---

> > > ### Author Response · Authors · 2025-08-06
> > > **Response to Reviewer X5JK.**
> > >
> > > > **1. Practical Utility under Extreme Noise (when $\Delta$ MPJPE > 100)**
> > >
> > > It's true that when $\Delta$ MPJPE exceeds 100, the results are poor for both models in terms of accuracy. However, there is a meaningful difference in **failure behavior** between the baseline and our method under such extreme noise levels.
> > >
> > > - **Baseline** suffers from a catastrophic failure, producing physically implausible outputs that break the skeletal structure under high perturbation. This can cause fatal errors (e.g., undefined behavior) in applications like robotics, making post-processing difficult or impossible.
> > > - **Our method**, on the other hand, avoids this catastrophic failure, maintaining a "inaccurate but plausible" human-like structure in our experiment.
> > >
> > > |Model|0|0.5|1|1.5|2|2.5|3|5|
> > > |-|-|-|-|-|-|-|-|-|
> > > |BaselineNet (MPJPE &darr;)|89.8|347.8|646.5|969.0|1291.3|1632.1|1981.1|3345.0|
> > > |BaselineNet+Ours (MPJPE &darr;)|**85.6**|**251.3**|**301.5**|**321.8**|**331.9**|**339.9**|**342.7**|**352.1**|
> > >
> > > This resilience is crucial in real-world scenarios, where factors like lighting and extreme occlusion often cause high input noise. It allows fail-safe mechanisms (e.g., robot system can still identify a human shape in a danger zone, even if the predicted pose is inaccurate) and enables effective post-processing. In essence, our method ensures stability and prevents catastrophic failures under high noise levels, where other models break down.
> > >
> > > > **2. Correction for Figure 6(d)**
> > >
> > > Thank you for pointing out the issue in Fig. 6(d). We mistakenly showed results from Human3.6M test set instead of 3DHP, contrary to our intention in Section 4.7. The correct results for 3DHP are provided in the rebuttal table above. We note that both datasets show similar tendency. We appreciate the opportunity to clarify this and will update the figure accordingly in the future version.

---

### Official Review · Reviewer_s6cb · 2025-06-19

**Clarity:** 3
**Significance:** 3
**Originality:** 3
**Rating:** 5
**Confidence:** 2

**Summary:**

This paper poses the critical issue of generalization in 3D Human Pose Estimation (HPE) by shifting focus from network architecture design to the loss landscape geometry. The authors first analyze the complex and fragmented loss landscapes present in HPE, particularly those arising from depth ambiguities. They introduce an adaptive scaling mechanism that encourages flatness in the local minima of the loss surface and propose an efficient ensembling framework based on multiple regression heads. This approach allows the model to explore and average diverse but high-performing solutions within a smoothed loss region. Experiments on Human3.6M and MPI-INF-3DHP demonstrate consistent improvements across multiple architectures (MLP, CNN, GCN, Transformer), both in-domain and cross-dataset.

**Questions:**

The paper focuses on 3D HPE. Could the proposed method be applied to other tasks with similarly complex loss landscapes, such as 2D pose estimation or object detection?

**Ethical Concerns:**

["NO or VERY MINOR ethics concerns only"]

**Final Justification:**

Thank you for your response. I will keep the original score.

**Limitations:**

Yes.

**Quality:**

3

**Strengths And Weaknesses:**

1. Novel perspective on generalization in 3D HPE: The paper introduces an underexplored but highly relevant perspective by analyzing the loss landscape of 3D HPE. The findings are well-motivated and supported by visualizations and eigenvalue analysis.
2. Simple yet Effective Adaptive Scaling: The adaptive scaling mechanism is elegant, architecture-agnostic, and incurs minimal additional computational cost.
3. Thorough Experimental Validation: The paper conducts extensive experiments with multiple architectures and benchmarks, including cross-dataset evaluations, noise robustness, and comparison with SAM. Ablation studies clearly isolate the contribution of each component.

---

> ### Author Rebuttal · Authors · 2025-07-31
>
> We thank the reviewer for raising this insightful question regarding the applicability of our method beyond 3D human pose estimation.
>
> > # Q1. The possibility of applying our method to other tasks.
>
> While our approach is tailored to the 3D HPE specific challenge, we believe the underlying principle, “encouraging local flatness in the loss landscape”, could potentially be beneficial for other tasks that suffer from similar optimization difficulties due to complex or multimodal loss surfaces. However, successful adaptation would likely require careful consideration of task-specific factors. For instance, the location or strategy of applying adaptive scaling might need to be modified depending on the architecture and characteristics of the target task. So, further investigation is needed, and we leave this as an exciting direction for future work.

---

### Official Review · Reviewer_MPr4 · 2025-06-28

**Clarity:** 3
**Significance:** 3
**Originality:** 3
**Rating:** 5
**Confidence:** 3

**Summary:**

This paper addresses the challenge of generalization in 3D Human Pose Estimation (HPE). The authors argue that the loss landscape of 3D HPE is highly complex and has multiple disconnected local minima, which can negatively impact model generalization. To address this, they propose a simple adaptive scaling mechanism that smooths the loss landscape, reducing the likelihood of high-loss barriers between local minima. Furthermore, they introduce an efficient ensemble approach that combines diverse solutions on the smoothed loss landscape. The method is designed to be model-agnostic and easy to implement. Experimental results on benchmark datasets such as Human3.6M and MPI-INF-3DHP demonstrate improvements in generalization performance across various network architectures.

**Questions:**

1. How do you ensure the diversity of solutions when using different random seeds? Is there a risk of the models converging to similar local minima despite different initialization?

2. Can you provide more insights into how the adaptive scaling mechanism affects the model's learning process? Are there any potential drawbacks or limitations to this mechanism?

**Ethical Concerns:**

["NO or VERY MINOR ethics concerns only"]

**Final Justification:**

The rebuttal addressed most of my concerns. I would keep my original rating.

**Limitations:**

Yes, but it is in the supplementary material.

**Quality:**

3

**Strengths And Weaknesses:**

Strengths:

1. The paper provides a novel analysis of the 3D HPE task from a loss landscape perspective, which is underexplored in previous research. This analysis offers valuable insights into the challenges of optimizing 3D HPE models and their generalization performance.

2. The proposed adaptive scaling mechanism and ensemble approach are simple yet effective. They are designed to induce flatness in the loss landscape, which is associated with better generalization. The method shows consistent performance improvements across different model architectures and datasets.

3. The experiments are comprehensive and well-designed. The authors demonstrate the effectiveness of their method on multiple benchmark datasets and compare it with various state-of-the-art architectures. The results show significant improvements in generalization performance, validating the practical utility of the proposed approach.

4. The method can be easily integrated into existing 3D HPE frameworks regardless of the model architecture. This makes it a versatile enhancement for the community.


Weaknesses:

1. While the paper provides empirical evidence of the effectiveness of the proposed method, the theoretical understanding of why and how the adaptive scaling mechanism and ensemble approach work is somewhat limited. More in-depth theoretical analysis could strengthen the paper.

2. The paper primarily focuses on the generalization performance within the context of benchmark datasets. It would be beneficial to discuss the applicability of the method in real-world scenarios where data distributions may vary more significantly.

---

> ### Author Rebuttal · Authors · 2025-07-31
>
> We sincerely thank the reviewer for the positive comments and for recognizing the value of our contributions. We truly appreciate the careful reading and constructive feedback. Below, we provide detailed responses to each of the reviewer’s points.
>
> > # W1. A simple theoretical understanding of our method.
>
> We provide a simple theoretical understanding of our method below.
>
> - **For Adaptive Scaling Mechanism**,
>
>
> The formulation of Eq. 3 in the main paper is $\frac{f_{\theta_i}(X)}{\sigma(h_{\psi_i}(X))+1}$, where $\sigma$ is ReLU function, $g_\phi(x) = X$, and $x$ is input.
>
> And let’s say that $\frac{f_{\theta_i}(X)}{\sigma(h_{\psi_i}(X))+1} = \frac{f_{\theta_j}(X)}{\sigma(h_{\psi_j}(X))+1} = \tilde{y}$. So each head has same output and same loss value.
>
> When $\sigma(h_{\psi_i}(X))+1 = a$ and  $\sigma(h_{\psi_j}(X))+1 = a+\Delta a$, then $f_{\theta_i}(X) = a\tilde{y}$ and $f_{\theta_j}(X) = a\tilde{y} + \Delta a\tilde{y}$. ($a$ is positive)
>
> By substitution, $\sigma(h_{\psi_j}(X)) - \sigma(h_{\psi_i}(X)) = \Delta a$ and $f_{\theta_j}(X) - f_{\theta_i}(X) = \Delta a\tilde{y}$.
>
> By the Taylor expansion, $f_{\theta_j}(X) \approx  f_{\theta_i}(X) + \nabla_\theta f_{\theta_i}(X)^T (\theta_j - \theta_i)$.
> Similary, $\sigma(h_{\psi_j}(X)) \approx \sigma(h_{\psi_i}(X)) + \nabla_\psi \sigma(h_{\psi_i}(X))^T (\psi_j - \psi_i)$.
>
> For simplicity, we address the case of $\theta$ here.
>
> By the substitution, $\nabla_\theta f_{\theta_i}(X)^T (\theta_j - \theta_i) \approx \Delta a\tilde{y}$.
>
> Here, the gradient $\nabla_\theta f_{\theta_i}(X)$ has large enough rank and is non-zero in practical.
>
> So when $\Delta a \rightarrow 0$, the $(\theta_j - \theta_i) \rightarrow 0$. This implies that there exists connected region in $\theta$ space where shares same loss value depending on the value of $\Delta a$.
>
>
>
> - **For Ensemble strategy**,
>
>
> Let’s say that $S_j = S_i + \Delta S_{ij}$, where $S_i = (\phi, \theta_i, \psi_i)$.
>
> By Taylor expansion of  $F_{S_j}(x) = y,   F_{S_i+\Delta S_{ij}} \approx F_{S_i}(x) + \nabla_S F_{S_i}(x)^T \Delta S_{ij}$.
>
> If $\nabla_S F_{S_i}(x)^T$ and $\Delta S_{ij}$ are non-zero, then $F_{S_i}(x) \neq F_{S_j}(x)$, which means they are different function.
> In practical, the gradient is not zero vector.
> And to make $\Delta S_{ij} \neq 0$, we intentionally initialized each head differently as mentioned in the paper.
>
> As a result, each $F_{S_i}(x)$ is different functions so this can justify our ensemble strategy.
>
>
> > # W2. Real-world scenario (more diverse dataset).
>
> we trained on a combined dataset (H36M + 3DHP + SkiPose), where SkiPose adds dynamic, realistic postures, better reflecting real-world scenarios.
> For details, we use GT 2D pose input for all model and set the number of heads as 3.
> Our experiments show consistent improvements across metrics on 3DPW, confirming that our method benefits from and effectively handles increased data diversity.
>
> |Model|MPJPE &darr;|PA-MPJPE &darr;|PCK &uarr;|
> |-|-|-|-|
> |BaselineNet|92.2|57.5|84.1|
> |BaselineNet+Ours|**85.9**|**55.8**|**85.9**|
> |VPose|93.6|59.6|84.3|
> |VPose+Ours|**87.1**|**56.4**|**85.8**|
>
> We can explain this as following. Since 3D HPE is inherently a one-to-many problem due to depth ambiguity, a single input can be matched to multiple solutions (the model can have multiple valid loss values for a single input). As a result, the resulting loss landscape becomes highly complex as data diversity increases. So our approach that smooths such complex landscapes can work well for diverse dataset.
>
> > # Q1. Ensuring the diversity of solutions from each head.
>
> We address this question through a simple theoretical understanding.
>
> Let’s say $F_i(x) = \frac{f_{\theta_i}(g_\phi(x))}{\sigma(h_{\psi_i}(g_\phi(x)))+1}$ and $L=\frac{1}{M}\sum_{i=1}^{M}L_i(F_i(x),y)$, where $M$ is the number of heads and $L_i$ is loss value of $i^{th}$ head.
> Then, $\frac{\delta L}{\delta \theta_i} = \frac{1}{M}\frac{\delta L_i}{\delta F_i}\frac{\delta F_i}{\delta f_i}\frac{\delta f_i}{\delta \theta_i} + \alpha$.
> Similary, $\frac{\delta L}{\delta \psi_i} = \frac{1}{M}\frac{\delta L_i}{\delta F_i}\frac{\delta F_i}{\delta h_i}\frac{\delta h_i}{\delta \psi_i} + \beta$.
>
> At the start of training, we intentionally initialized each head differently in weight scale. This means that $F_i(x) \neq F_j(x)$ and $L_i \neq L_j$ at the start of training.
> Thus, $\frac{\delta L}{\delta \theta_i} \neq \frac{\delta L}{\delta \theta_j}$ at the start of training, which means each head get different loss gradient from the start of training. This also holds in the case of $\psi$.
>
> However, while the possibility is low, there is no rigorous guarantee that each head will always converge to different local minima. For this reason, as mentioned in the paper, we intentionally initialized each head with different parameter values to encourage diversity in their convergence behavior.
>
> > # Q2. Insight for the impact of adaptive scaling mechansim on the learning process of the model.
>
> As we mentioned in the paper, the adaptive scaling mechanism smooth the loss landscape, which means that the optimization becomes easier than before.
> This means that the adaptive scaling mechanism accelerates convergence. So, the early stopping may be needed when applying the adpative scaling mechansim to avoid overfitting.
> In fact, when we investigate the loss trajectories w/wo adaptive scaling mechanism only through the epochs, we can see the adaptive scaling mechanism boost the converging speed as shown in below table.
> Note that E1, E2, … refer to the epoch.
>
> |Model|E1|E2|E3|E4|E5|E6|E7|E8|E9|E10|
> |-|-|-|-|-|-|-|-|-|-|-|
> |BaselineNet (Loss)|0.1137|0.0084|0.0045|0.0032|0.0025|0.0021|0.0018|0.00165|0.0014|0.0013|
> |BaselineNet+Scale only (Loss)|**0.0172**|**0.0031**|**0.0023**|**0.0019**|**0.0016**|**0.0014**|**0.0013**|**0.0012**|**0.0011**|**0.0010**|
> |BaselineNet (MPJPE)|115.8|92.1|91.3|74.6|66.1|61.2|60.6|58.5|60.2|59.5|
> |BaselineNet+Scale only (MPJPE)|**71.9**|**63.1**|**59.4**|**57.3**|**56.4**|**56.5**|**55.3**|**55.4**|**55.3**|**54.8**|

---

### Official Review · Reviewer_8o6Y · 2025-07-04

**Clarity:** 3
**Significance:** 2
**Originality:** 3
**Rating:** 4
**Confidence:** 3

**Summary:**

This work presents a training and ensemble strategy for 3D human pose estimation (3D HPE) based on loss landscape analysis. The motivation is that the 3D HPE’s loss surface contains multiple disconnected sharp minima that hinder generalization. To address this problem, an adaptive scaling mechanism is adopted to smooth the loss landscape and encourage the discovery of diverse, generalizable solutions. Building on this, they propose a simple and architecture-agnostic ensemble method that effectively leverages solution diversity to enhance robustness, all with minimal training overhead. Experiments are conducted on Human3.6M and MPI-INF-3DHP, to validate the effectiveness of the proposed designs on 4 typical architectures, MLP, CNN, GCN, and Transformer. The proposed designs achieve promising improvement (over 1.6% at most time) in both MPJPE and PA-MPJPE.

**Questions:**

1. Please add the details of experiment settings.
2. Experiments that train with more datasets would be very helpful to identify the significance of the proposed method.

**Ethical Concerns:**

["NO or VERY MINOR ethics concerns only"]

**Final Justification:**

Via training with an additional in-the-wild dataset, SkiPose, and test on 3DPW, the proposed method still achieve promising improvement than the baseline model. Such an experiment may indicate that the proposed method could be beneficial to models trained on large-scale and diverse datasets. Different from BEDLAM, SkiPose is relatively limited in diversity of human motion. Please consider to show the results using BEDLAM in the final version. Because we still can't be sure of how much improvement the proposed method can bring to the models trained with large-scale and diverse datasets, the significance of the proposed method is hard to determine in its current state, therefore, I keep the original score.

**Limitations:**

Yes.

**Paper Formatting Concerns:**

No.

**Quality:**

2

**Strengths And Weaknesses:**

Strengths.
1. New perspective.
This paper introduces a insightful approach by analyzing the 3D HPE at the perspective of loss landscape, which is rarely explored in this field.
2. Generalizable design.
The proposed adaptive scaling mechanism seems to be easy to added to different model and architecture-agnostic. This great property makes it widely applicable. Experiments on Human3.6M and MPI-INF-3DHP validate it.
3. Insightful experiments.
This paper provides insightful experiments to show that the proposed method leads to flatter local minima by estimating the top-1 eigenvalue of the Hessian matrix and visualizing both local and global loss landscapes. This analysis shows that adaptive scaling reduces curvature and eliminates high-loss barriers between solutions, which supports the claim of improved generalization.
Also, the comparison with Sharpness-Aware Minimization (SAM), an existing method with global flattening effect, shows that the proposed   method can avoid biasing the landscape around specific local minimal.


Weakness.
1. Lacking sufficient details of experiment settings.
To further convince the readers, we really need to add more details of the experiments on H36M and MPI-INF-3DHP, including the training datasets used in Tab. 3 and 4, how to combine the SAM, etc. The SAM comparison would benefit from stronger experimental support, such as better tuning or combining with ensembling.
Also, the Hessian estimation method (e.g., power iteration, approximation details) is not clearly described and should be explicitly stated for reproducibility.
2. The models are only trained on Human3.6M. Whether the proposed can help the model trained on more datasets remains unclear.
One potential limitation is that all experiments are conducted with models trained exclusively on the Human3.6M dataset. While the cross-dataset evaluation on MPI-INF-3DHP is valuable, it remains unclear whether the proposed adaptive scaling mechanism can provide similar benefits in more realistic settings where models are trained on large-scale and diverse datasets (e.g., combined H36M + MPI-INF-3DHP + BEDLAM, and validate on 3DPW). In such settings, the optimization landscape may already exhibit flatter or more entangled structures, potentially reducing the marginal gain from the proposed smoothing strategy. The generality of the method in large-data regimes thus remains an open question and could be further explored.

---

> ### Author Rebuttal · Authors · 2025-07-31
>
> We sincerely thank the reviewer for recognizing the strengths of our work, especially our loss landscape perspective, generalizable design, and insightful analysis. We appreciate the constructive feedback and provide the following clarifications:
>
> > # W1. More details of experimental settings.
>
> - **Dataset & Training**: Tab. 4 and 5 use Human3.6M (Train: S1,5,6,7,8 / Test: S9,11) without flip augmentation. All models in Tab. 4 and 5 are trained for 60 epochs due to the slow convergence, especially under SAM and ensemble settings.
> - **SAM Settings**: We used perturbation radius = 0.05 in Tab. 3. Below are results for various radiuses, with and without our ensembling strategy (3 heads):
>
>
> |Model|0.01|0.02|0.03|0.05|0.07|
> |-|-|-|-|-|-|
> |BaselineNet + SAM|55.0|55.5|55.8|56.1|56.5|
> |BaselineNet + SAM + Ens.|**54.2**|**54.4**|**54.5**|**54.5**|**54.6**|
> |VPose + SAM|56.3|56.7|57.1|57.3|57.8|
> |VPose + SAM + Ens.|**55.0**|**55.1**|**55.3**|**55.3**|**55.5**|
>
> As we expected, the performances are not satisfactory with various perturbation radii of SAM.
> However, SAM + Ensemble shows some performance gains over SAM alone, as compared to Table 4 in the main paper. This supports our claim that the loss landscape of 3D HPE is highly complex, with diverse local minima that can be better exploited via ensembling.
>
> Also, comparing to our method (BaselineNet+Scale: **54.2**, BaselineNet+Both: **53.6**, VPose+Scale: **54.9**, and VPose+Both:**54.4**), the results show that our method outperforms SAM.
>
> - **Hessian estimation**: We use the power iteration method with maximum 100 steps and a tolerance of 0.001 for top-1 eigenvalue computation.
>
> > # W2. The effect on more diverse training dataset.
>
> We train our method on a combined dataset (H36M+3DHP+SkiPose) and evaluate the model on 3DPW. Note that we try to download BEDLAM, however, downloading BEDLAM takes a couple of days. In this reason, we leverage SkiPose instead of BEDLAM.
> SkiPose provides dynamic motion diversity, which reflects diverse training data distribution.
> For details, we use GT 2D pose input for all model and set the number of heads as 3.
> The results show our method can be more effective on diverse dataset, which supports our motivation.
>
> |Model|MPJPE &darr;|PA-MPJPE &darr;|PCK &uarr;|
> |-|-|-|-|
> |BaselineNet|92.2|57.5|84.1|
> |BaselineNet+Ours|**85.9**|**55.8**|**85.9**|
> |VPose|93.6|59.6|84.3|
> |VPose+Ours|**87.1**|**56.4**|**85.8**|
>
> We explain this results as below.
>
> 3D HPE is an ill-posed problem — a single 2D input can correspond to multiple plausible 3D poses. This implies that, at a given parameter point $\theta$, a single input can induce multiple possible loss values due to depth ambiguity. We can quantify this ambiguity using the Depth Ambiguity Ratio (DAR). The higher DAR leads to greater variance in possible loss values, which in turn suggests sharper or more irregular loss landscapes. This is empirically supported by Fig. 2(b) in the paper. Therefore, when training on more diverse datasets (which often have a higher range of DAR), the resulting loss landscape becomes more complex and fluctuating.
>
> In fact, by examining the top-1 eigenvalue of the Hessian matrix for each $C_k$ in the combined dataset, we can see that the local loss landscape geometry becomes significantly sharper compared to the model trained only on Human3.6M.
> Below are results when we set K as 21. For hessian estimation, we set power iteration as 100 and tolerance as 0.001.
>
> |Class|$C_1$|$C_2$|$C_3$|$C_4$|$C_5$|$C_6$|$C_7$|$C_8$|$C_9$|$C_{10}$|$C_{11}$|$C_{12}$|$C_{13}$|$C_{14}$|$C_{15}$|$C_{16}$|$C_{17}$|$C_{18}$|$C_{19}$|$C_{20}$|$C_{21}$|
> |-|-|-|-|-|-|-|-|-|-|-|-|-|-|-|-|-|-|-|-|-|-|
> |$\lambda_{max}$(H36M)|2.0|1.4|1.3|1.2|1.1|1.1|1.1|1.1|1.1|1.1|1.2|1.3|1.5|1.8|2.2|3.0|3.6|4.0|4.0|4.4|8.3|
> |$\lambda_{max}$(combined dt.)|**3.8**|**3.3**|**2.9**|**2.6**|**2.5**|**2.4**|**2.4**|**2.3**|**2.3**|**2.4**|**2.6**|**3.0**|**3.7**|**4.7**|**6.4**|**8.1**|**9.3**|**16.2**|**67.8**|**60.1**|**62.8**|

---

> > ### Comment · Reviewer_8o6Y · 2025-08-07
> >
> > Thanks for your efforts.
> > Via training with an additional in-the-wild dataset, SkiPose, and test on 3DPW, the proposed method still achieves promising improvement than the baseline model. Such an experiment may indicate that the proposed method could be beneficial to models trained on large-scale and diverse datasets.
> > Different from BEDLAM, SkiPose is relatively limited in diversity of human motion. Please consider to show the results using BEDLAM in the final version.
> > Because we still can't be sure of how much improvement the proposed method can bring to the models trained with large-scale and diverse datasets, the significance of the proposed method is hard to determine.

---

> > > ### Author Response · Authors · 2025-08-08
> > > **Thank you for your thoughtful discussion.**
> > >
> > > We sincerely thank you for your valuable feedback and thoughtful discussion. While BEDLAM experiments could not be included in the rebuttal due to time constraints, our current findings suggest that the proposed method is likely to perform well on BEDLAM. As you suggested, we will include these experiments in the final version to further validate the significance of the proposed method. We appreciate your time and consideration.

---

### Note · Authors · 2025-08-13

We thank the reviewers and Area Chair for their time and insightful feedback. To aid the final assessment, we summarize our paper's acknowledged strengths and our responses to key concerns.

># **Acknowledged Strengths**

A strong consensus emerged on several key merits:
- **Novel and Insightful Perspective**: Its novel analysis of the loss landscape geometry in 3D HPE was recognized as a key contribution (8o6Y, MPr4, s6cb).
- **Simple and Effective Method**: The method's elegance, simplicity, and effectiveness in enhancing generalization were praised (MPr4, s6cb, X5JK).
- **Broad Applicability**: Its generalizability and ease of integration into various architectures were highlighted as a major asset (8o6Y, MPr4).
- **Insightful Experiments**: The comprehensive and well-designed experiments were commended for clearly demonstrating the method's effectiveness (8o6Y, MPr4, s6cb).

># **Addressed Key Concerns**

We have diligently addressed all concerns via detailed clarifications and new experiments.

**Shared Concerns**
- **Generalization to Diverse Datasets** (8o6Y, MPr4): New experiments on a combined dataset (H36M+3DHP+SkiPose) demonstrated our method's efficacy on 3DPW. We will add the requested BEDLAM experiments in the final version.
- **Theoretical Explanation** (MPr4, X5JK): We provided a simple, yet intuitive, theoretical explanation for why our adaptive scaling mechanism works and how our ensemble strategy promotes diversity.

**Individual Concerns**
- **Connection to 3D HPE** (sYww): Our clarification on the motivation from 3D HPE's inherent depth ambiguity was well-received, resolving their primary concern.
- **Practicality under high noise** (X5JK): The additional experiment shows that a distinguishing advantage of our method is its crucial real-world resilience: avoiding catastrophic failures (e.g., implausible poses) under high noise, unlike the baseline.
- **Analysis of Varying K Values** (X5JK): Our additional analysis showed that our core findings are not sensitive to varying K values, supporting our design choice.

># **Final Summary**

Our work provides a new lens for the generalization problem in 3D HPE: **the perspective of its underlying loss landscape**. By identifying and tackling underexplored 3D HPE optimization difficulties arising from the task's inherent ambiguities, we developed a simple and effective framework. We will incorporate all feedback into the final version and thank you again for your consideration.

---

### Decision · Program_Chairs · 2025-09-17

**Decision:**

Accept (poster)

**Comment:**

This paper addresses the generalization challenge in 3D Human Pose Estimation by analysing the geometry of its loss landscape. The paper proposes an adaptive scaling mechanism to smooth the loss surface, enabling models to find more diverse and generalisable solutions. Building on this, the paper proposes a simple ensemble strategy that effectively leverage this diversity. Experiments on Human3.6M and MPI-INF-3DHP show consistent performance gains across multiple architectures.

Reviewers initially raised several concerns including the unclear relationship between the proposed method and 3D HPE, the lack of experiments supporting generalisation to the diverse datasets, and limited theoretical justification of the adaptive scaling mechanism and ensemble approach. During the rebuttal, the authors addressed these concerns satisfactorily, and as a result all reviewers become generally positive about this paper. However, it is important that the requested revisions are incorporated into the final version, in particular showing the results using BEDLAM as noted by reviewer 8o6Y.

Based on the overall positive feedback, the AC recommends acceptance.